# Promising Oldest Ice sites in East Antarctica based on thermodynamical modelling

Brice Van Liefferinge[1], Frank Pattyn[1], Marie G.P. Cavitte[2,3], Nanna B. Karlsson[4,*], Duncan A. Young[2], Johannes Sutter[4,5], and Olaf Eisen[4,6]

[1]Laboratoire de Glaciologie, Université libre de Bruxelles, CP 160/03, Avenue F.D. Roosevelt 50, B-1050 Brussels, Belgium
[2]Institute for Geophysics, Jackson School of Geosciences, University of Texas at Austin, Austin, Texas, USA
[3]Department of Geological Sciences, Jackson School of Geosciences, University of Texas at Austin, Austin, Texas, USA
[4]Alfred Wegener Institute Helmholtz-Centre for Polar and Marine Research, D-27568 Bremerhaven, Germany
[5]Climate and Environmental Physics, Physics Institute, and Oeschger Centre for Climate Change Research, University of Bern, Bern, Switzerland
[6]Department of Geosciences, University of Bremen, Germany
[*]now at: Geological Survey of Denmark and Greenland (GEUS), Oster Voldgade 10, 1350 Copenhagen, Denmark

*Correspondence to:* Brice Van Liefferinge
(bvlieffe@ulb.ac.be)

**Abstract.** To resolve the mechanisms behind the major climate reorganisation which occurred between 0.9 and 1.2 Ma, the recovery of a suitable 1.5 million-year-old ice core is fundamental. The quest for such an Oldest Ice core requires a number of key boundary conditions, of which the poorly known basal geothermal heat flux (GHF) is lacking. We use a transient thermodynamical 1D vertical model that solves for the rate of change of temperature in the vertical, with surface temperature and modelled GHF as boundary conditions. For each point on the ice sheet, the model is forced with variations in atmospheric conditions over the last 2 Ma, and modelled ice-thickness variations. The process is repeated for a range of GHF values to determine the value of GHF that marks the limit between frozen and melting conditions over the whole ice sheet, taking into account 2 Ma of climate history. These threshold values of GHF are statistically compared to existing GHF data sets. The new probabilistic GHF fields obtained for the ice sheet thus provide the missing boundary conditions in the search for Oldest Ice. High spatial resolution radar data are examined locally in the Dome Fuji and Dome C regions, as these represent the ice core community's primary drilling sites. GHF, bedrock variability, ice thickness and other essential criteria combined highlight a dozen major potential Oldest Ice sites in the vicinity of Dome Fuji and Dome C, where GHF allows for Oldest Ice.

## 1 Introduction

The relationship between the variations in atmospheric $CO_2$ and atmospheric temperatures, determined from oxygen isotope records, is increasingly better understood through a wealth of marine and lacustrine records recently recovered (Kawamura et al., 2017). However, characterising this relationship on short time scales, with direct sampling of the paleo-atmosphere, requires a temporal resolution that can only be obtained from ice core records, which currently only go back as far as 800 ka (EPICA community members, 2004; Parrenin et al., 2007). In particular, there is a strong interest in constraining greenhouse

gas forcings between 0.9 and 1.2 Ma, a period during which glacial-interglacial periodicity changed from 40 ka to 100 ka cycles (the so called Mid-Pleistocene Transition or MPT, Lisiecki and Raymo, 2005; Snyder, 2016) but without explained natural forcings (e.g. Milankovitch, regolith base, size of the ice sheet; Imbrie, 1993; Clark et al., 2006; Elderfield et al., 2012). To resolve the mechanisms behind the major climate reorganisation during the MPT, the recovery of suitable 1.5 million-year-

old ice core samples is fundamental. Such old ice would provide us with unique and crucial insights into air composition as well as the isotopic composition and dust content of the ice throughout the MPT.

    In order to retrieve a 1.5 million-year-old ice core in the center of the Antarctic Ice Sheet (so called Oldest Ice, Wolff et al., 2005), the base of the ice sheet should not have experienced melting or refreezing processes during this period (Wolff et al., 2005; Fischer et al., 2013). Furthermore, even in regions where basal melting can be considered to be insignificant, complex

processes of mixing or folding due to rough bedrock topography can cause perturbations in ice flow over the bedrock and make accurate dating of the ice difficult or even impossible (Bell et al., 2011). These processes have impacted the NEEM ice core analysis in Greenland (Dahl-Jensen et al., 2013) as well as the signal of the deeper part of the EPICA Dome C ice core (Tison et al., 2015). Moreover, in order to recover an interpretable climate signal, present-day ice surface velocities should remain below a certain threshold (less than 1 to 2 m a$^{-1}$ for the horizontal surface velocities), so that ice has travelled as little as possible

horizontally. Finally, ice should be as thick as possible in order to preserve a resolvable and thus an interpretable record within the deeper layers. In 2013, Fischer et al. (2013) and Van Liefferinge and Pattyn (2013) evaluated the conditions necessary for retrieving an old ice core record and highlighted candidate sites with potential 1.5 million-year-old ice in Antarctica. They stressed the importance of collecting denser ice thickness coverage over such candidate sites to reduce uncertainties in the modelled basal ages and basal temperature conditions.

The major uncertainty in determining basal melting and basal temperature gradients stems from our limited knowledge of the spatial distribution of the geothermal heat flux (GHF) underneath the Antarctic Ice Sheet. As direct measurements are challenging, due to the presence of the thick ice cover, several approaches exist to derive GHF distributions based on limited data (Shapiro and Ritzwoller, 2004; Fox-Maule et al., 2005; Purucker, 2013; An et al., 2015; Martos et al., 2017). All methods infer GHF from properties of the crust and the upper mantle and therefore provide average GHF values without accounting

for crustal gradients in GHF. Furthermore, from an ice-sheet modelling perspective, it is crucial to know basal temperature gradients at the ice-bedrock interface and not GHF within the crust. Shapiro and Ritzwoller (2004) extrapolated the GHF from a global seismic model of the crust and the upper mantle. Fox-Maule et al. (2005) derived the GHF from satellite magnetic measurements, and Purucker (2013) provided a GHF data set as an update of the latter. More recently, An et al. (2015) analysed the Earth's mantle properties from a new 3D crustal shear velocity model to calculate crustal temperature and surface GHF.

Their GHF values for East Antarctica deviate by $\pm 10$ mW m$^{-2}$ compared to Shapiro and Ritzwoller (2004), which used a similar method. They found very low GHF values, $\sim$40 mW m$^{-2}$, in areas close to Dome C, Dome Fuji and Dome Argus, as well as across the Gamburtsev subglacial mountains. Their model, however, is invalid for GHF exceeding 90 mW m$^{-2}$, but these high values concern the young areas of the crust, mainly in West Antarctica and the Transantarctic mountains. Finally, Martos et al. (2017) provided the first high resolution heat flux map on a 15 km by 15 km grid derived from the spectral analysis of a

continental compilation of airborne magnetic data. Generally low values of GHF are found in East Antarctica with respect to

West Antarctica. This data set estimates the GHF across all candidate sites with variations of up to 20 % from other data sets (Dome F ($65 \pm 12$ mW m$^{-2}$), Dome C ($58 \pm 12$ mW m$^{-2}$) and Dome A ($55 \pm 11$ mW m$^{-2}$)). The five data sets differ both in absolute values as well as in their spatial distribution of GHF.

On smaller spatial scales, those particularly relevant to the search for Oldest Ice, GHF is constrained by using models based on ice-penetrating radar, on scales ranging from 1 km$^2$ to 100 km$^2$ (Parrenin et al., 2017; Passalacqua et al., 2017). Spatially localized features such as lakes and deep ice-core drillings have to be taken into account when attempting to constrain GHF. Subglacial lakes have been documented under the ice of the Antarctic Ice Sheet through the collection of radar and seismic data. An ever increasing number of lakes have been identified and the current count is close to 415 (Smith et al., 2009; Wright and Siegert, 2012; Young et al., 2016). However, more lakes are suspected to exist in currently unsurveyed areas. With respect to deep ice core drill sites, only a few drillings have reached the actual ice-bedrock interface, enabling a direct measurement of the GHF (or at least the basal temperature gradient), i.e. Vostok (Petit et al., 1999; Parrenin et al., 2004), EPICA Dome C (EPICA community members, 2004; Parrenin et al., 2007), Dome Fuji (Fujii et al., 2002; Hondoh et al., 2002; Watanabe et al., 2003), and EPICA Dronning Maud Land (EPICA community members, 2006; Ruth et al., 2007). All drillings revealed a basal temperature close to or at pressure melting point (pmp).

Since the initial efforts to identify areas of 1.5 million-year-old-ice sites (Fischer et al., 2013), a lot of progress has been made in predicting such candidate sites through the collection of detailed ice-penetrating radar data (Steinhage et al., 2013; Cavitte et al., 2016). Models focussing on divide-adjacent areas and using these radar data also add confidence in the probability of detecting Oldest Ice (Parrenin et al., 2017). Furthermore, the mechanisms that control the geometry and the ice volume as well as Antarctic Ice Sheet stability are also increasingly better understood (Shakun et al., 2015; Pollard et al., 2015). Shakun et al. (2015) put forward the strong coupling between ice volume and temperature over climatic cycles from planktonic $\delta^{18}$O records. Pollard et al. (2015) put forward new mechanisms of hydrofracturing and ice cliff failure producing a rapid retreat of the ice sheet during past warm periods.

Dense ice-penetrating radar data recently collected over Dome Fuji and Dome C have been instrumental, not only to better constrain the most promising candidate Oldest Ice sites, but also to eliminate some of the modelled candidate sites. Processes active at the base of the ice sheet visible from radar data reduce chances of recovering Oldest Ice. This is the case for areas where significant subglacial water networks have been observed, such as seen in the vicinity of Dome Argus (Wolovick et al., 2013), or where subglacial lakes or subglacial trenches have been detected (Wright and Siegert, 2012; Young et al., 2016). Since the aim is to avoid melting at the base while preserving a sufficient ice-core resolution close to the basal-ice layers, this poses an additional problem: ice acts as an insulator, and therefore the greater the ice thickness, the warmer the ice at the base. However, thick ice is needed in order to sufficiently resolve the climatic signal at depth. Conversely, where freezing mechanisms (ice flow divergence, ridge-line freezing) or a reduced ice thickness prevent basal ice from melting, it has been shown that the probability for recovering Oldest Ice is greater, such as in the Gamburtsev Mountains (Creyts et al., 2014) or around Dome Fuji and Dome C (Young et al., 2017; Passalacqua et al., 2017). On a more detailed scale, Cavitte et al. (2018) have shown that near Dome C, the snow accumulation pattern is rather stable in time, leading to limited variations in surface topography over the last glacial cycles.

Obviously, the selection of candidate sites will be made building on radar data criteria. However, since our current radar coverage of the ice sheet interior is currently limited to small, localised areas, it is essential to use thermodynamic models to complement these radar data to characterize basal conditions. In addition, models have the advantage of highlighting areas of interest on small and large scales (Van Liefferinge and Pattyn, 2013; Pattyn, 2010; Passalacqua et al., 2017). In the context of the Oldest Ice initiative, recently collected radar data and modelling advances have highlighted three candidate areas in particular: Dome Fuji, Dome C as well as the Dome Argus area, even though radar data still needs to be refined for the latter (Wolovick et al., 2013; Sun et al., 2014). Furthermore, logistical issues cannot be ignored when deciding for the next deep ice core drilling site.

So far, thermomechanical modelling has been based on steady-state temperature fields (Van Liefferinge and Pattyn, 2013; Pattyn, 2010). However, previous interglacials were demonstrated to have had higher surface temperatures than today (Snyder, 2016; Lisiecki and Raymo, 2005), which, in combination with thicker ice (Pollard and DeConto, 2009; Pollard and Deconto, 2016) could impact basal temperatures and therefore the basal ice record. Given the size of the Antarctic Ice Sheet and the low vertical advection rates in the interior during prolonged glacial periods, steady-state conditions probably overestimate the probability of melting bed conditions. Here, we use a transient one-dimensional thermodynamical model to determine whether inland basal conditions over the last 1.5 million years remained frozen, and to determine in particular the threshold value of GHF ($G_{pmp}$) to satisfy these conditions. Our calculated threshold values are then statistically evaluated through a comparison with existing GHF datasets and their uncertainties (Fox-Maule et al., 2005; Purucker, 2013; Shapiro and Ritzwoller, 2004; An et al., 2015; Martos et al., 2017). The obtained probability distribution of ice that has remained frozen over the last 1.5 Ma is used to refine areas of potential Oldest Ice, both on a global scale, to re-examine previous distributions of potential Oldest Ice, and on a local scale to focus specifically on the Dome Fuji and Dome C districts using the new radar coverage (Karlsson et al., 2018; Young et al., 2017) and previous suggested constraints (Fischer et al., 2013).

## 2 Thermodynamical modelling

### 2.1 Steady-state model

Van Liefferinge and Pattyn (2013) analytically determined the minimum geothermal heat flux necessary to reach the pressure melting point at the base of the ice (Hindmarsh, 1999; Siegert, 2000). While a positive GHF increases the temperature at the base of the ice, the surface accumulation cools down the ice from the top. It follows that thick ice combined with a low accumulation requires a low GHF to avoid melting from occurring at the base. Although accumulation is relatively well constrained, this is not the case for GHF. In addition, available data sets (Shapiro and Ritzwoller, 2004; Fox-Maule et al., 2005; Purucker, 2013; An et al., 2015; Martos et al., 2017) have relatively large errors. In the vicinity of divide areas, GHF uncertainty is 55-70% and 40% for the Purucker (2013) and Fox-Maule et al. (2005) data sets and for the Shapiro and Ritzwoller (2004) data set, respectively, in our regions of interest. Van Liefferinge and Pattyn (2013) also used the GHF values from available data sets to calculate the basal temperature and highlight areas with basal melting by running a thermomechanical steady-state ice-flow model. The result was a map of mean basal temperatures on an ensemble model of 15 runs.

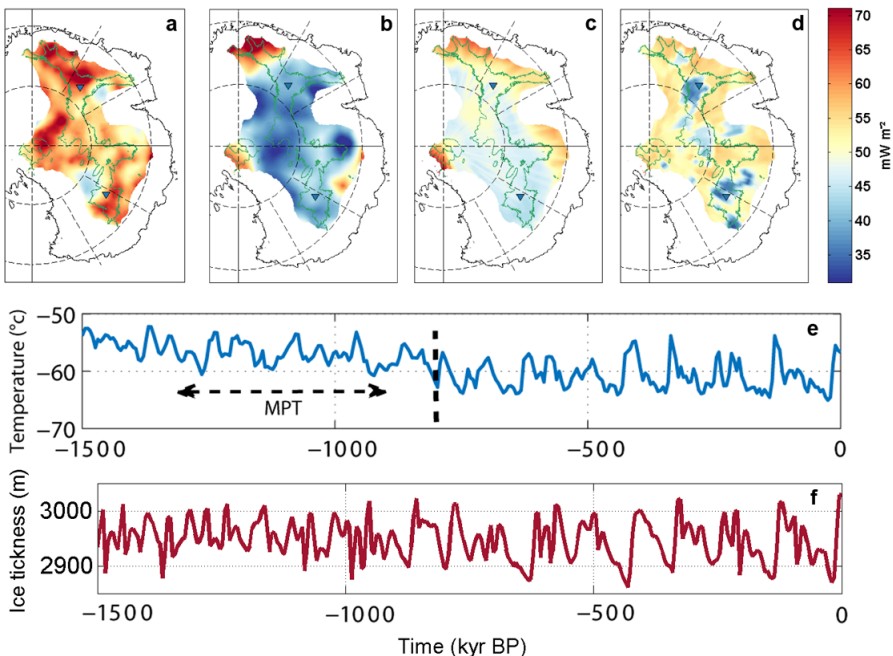

**Figure 1.** Top: GHF from A to D, Martos et al. (2017), Purucker (2013), Shapiro and Ritzwoller (2004) and An et al. (2015) data sets. GHF values are centred on the value of 51 mW m$^{-2}$. This value corresponds to the average of the GHF data sets within our area of interest. The green line outlines areas with surface velocities <2m a$^{-1}$ (calculated from balance velocities, Pattyn, 2010). GHF anomalies are limited to the 2700 m ASL surface elevation contour. The blue triangles locate Dome Fuji and Dome C ice cores (from top to bottom). Refer to Fig. 4 for the northing-easting polar stereographic grid and latitude-longitude coordinates. Bottom: E, Surface temperature (°C) reconstruction adapted from Snyder (2016) near Dome C. F, Ice thickness (m) reconstruction from Pollard and DeConto (2009) at Dome C.

## 2.2 Transient model description

In this paper we solve the vertical temperature profile over time, taking into account vertical diffusion and advection,

$$\frac{\partial T}{\partial t} = \frac{k}{\rho_i c_p} \frac{\partial^2 T}{\partial z^2} - w \frac{\partial T}{\partial z}, \tag{1}$$

where $k$ is the thermal conductivity, $\rho_i$ is ice density, $c_p$ is the heat capacity of ice, $T$ is the ice temperature, and $w$ the vertical ice velocity. Values for these parameters are listed in Table 1. In divide-adjacent areas, horizontal advection and horizontal heat conduction may be safely neglected as for areas with a relatively smooth bed, horizontal conduction is much lower than vertical conduction (Hindmarsh, 1999, 2018). Therefore the model is here limited to the interior slow-moving areas of the Antarctic Ice Sheet. In this perspective, the 2 m a$^{-1}$ surface velocity contour is highlighted on all the figures (derived from balance velocities; Pattyn, 2010) and used as the cut-off surface velocity for the search of Oldest Ice (Van Liefferinge and Pattyn, 2013). This

avoids the need for a correction of the climate signal due to upstream ice advection. The basal boundary condition for a cold base bed is given by

$$\frac{\partial T_b}{\partial z} = -\frac{G}{k}, \tag{2}$$

where $G$ is the geothermal heat flux. At the surface, the temperature is defined by a Dirichlet condition, i.e., $T = T_s$. The vertical velocity considers a profile based on simple shear using Glen's flow law with a flow exponent $n = 3$ (Hindmarsh, 1999; Pattyn, 2010),

$$w(\zeta) = -\dot{a}\frac{\zeta^{n+2} - 1 + (n+2)(1-\zeta)}{n+1}, \tag{3}$$

where $\zeta = (s - z)/h$, $h$ is the ice thickness, $s$ is the surface elevation, and $\dot{a}$ is the surface accumulation rate.

**Table 1.** Model parameters and constants.

| Symbol | Description | Unit | Value |
|--------|-------------|------|-------|
| $T_0$ | Absolute temperature | K | 273.15 |
| $k$ | Thermal conductivity | J m$^{-1}$ K$^{-1}$a$^{-1}$ | $6.627 \times 10^7$ |
| $G$ | Geothermal heat flux | W m$^{-2}$ | |
| $\rho$ | Ice density | kg m$^{-3}$ | 910 |
| $c_p$ | Heat capacity | J kg$^{-1}$ K$^{-1}$ | 2009 |
| $\gamma$ | Atmospheric lapse rate | K m$^{-1}$ | 0.008 |
| $n$ | Glen's flow law exponent | | 3 |

## 2.3 Model forcing

Atmospheric forcing is applied in a parameterized way, based on the observed fields of surface mass balance (accumulation rate) obtained from of the RACMO regional atmospheric climate model over the period 1980-2004, calibrated with observed surface mass balance rates (van de Berg et al., 2006; van den Broeke et al., 2006) and surface temperature (van den Broeke, 2008). For a change in background (forcing) temperature $\Delta T$, corresponding fields of precipitation $\dot{a}$ and atmospheric temperature $T_s$ are defined by Huybrechts et al. (1998) and Pollard and DeConto (2012).

$$T_s(t) = T_s^{\mathrm{obs}} - \gamma(s - s^{\mathrm{obs}}) + \Delta T(t), \tag{4}$$
$$\dot{a}(t) = \dot{a}^{\mathrm{obs}}\, 2^{(T_s(t) - T_s^{\mathrm{obs}})/\delta T}, \tag{5}$$

where $\gamma$ is the atmospheric lapse rate and $\delta T$ is 10°C (Pollard and DeConto, 2012). The subscript 'obs' refers to the present-day observed value. Any forcing (increase) in background then leads to an overall increase in surface temperature corrected for

elevation changes according to the environmental lapse rate $\gamma$. Surface elevation changes with time are obtained from changes in ice thickness with time obtained from a model that takes into account isostatic adjustment, given $s(t) = b + H(t)$, where $b$ is the bed elevation and $H(t)$ the time-varying ice thickness, defined by

$$H(t) = H_0 + (H^p(t) - H_0^p) , \tag{6}$$

where $H_0$ is the present-day ice thickness from Bedmap2 (Fretwell et al., 2013), updated with the local high resolution available data for Dome Fuji and Dome C (Karlsson et al., 2018; Young et al., 2017), and $H^p$ is the ice thickness variation in time obtained from ice sheet modelling over the last 2 Ma (Pollard and DeConto, 2009). Finally, background temperature changes $\Delta T(t)$ are taken from the reconstruction of Snyder (2016), discussed in the section 4.1, scaled to Dome C ice-core temperature reconstruction (Parrenin et al., 2007) (Fig. 1).

## 2.4 Limit values of GHF

The model is applied on a 5 km by 5 km grid size for the whole interior Antarctic Ice Sheet and on a 500 m by 500 m grid size for the two detailed analyses of Dome Fuji and Dome C, with 40 layers in the vertical. For each grid point within our Antarctic domain, the temperature profile is forced with changes in ice thickness, surface temperature and surface mass balance for a given GHF value. This is then repeated for a series of GHF values (Fig. 2), varying around a standard value of 51 mW m$^{-2}$.

We define $G_{pmp}$ as the threshold value of GHF for which basal melting may occur during the last 1.5 Ma. $G_{pmp}$ is determined using a 2nd degree polynomial fit function between GHF and the maximum basal temperature over the period of the last 1.5 Ma of each run as illustrated at the Dome C site in Fig. 3. GHF values that generate basal temperatures at the pressure melting point are not used for the fit function. To constrain the contribution of the ice cover to the basal temperature variation, the model is further run with uncertainties in the ice thickness. The chosen uncertainty corresponds to a 10% thicker and thinner ice sheet, which in our areas of interest is equivalent to a variation of 450 to 250 m in elevation. These differences are larger than the variations in ice thickness of the Pollard and DeConto (2009) reconstruction (between 50 and 250 m). A thicker ice cover insulates more than a thinner one and prevents heat flow from escaping as quickly to the surface. Our $G_{pmp}$ calculation indicates a variation of 6 to 8% for the threshold GHF due to the variation in ice thickness. For example, our value of the threshold GHF calculated at Dome C is 51.6 mW m$^{-2}$. With a higher and a thinner ice cover, these values reach 48.1 mW m$^{-2}$ and 55.9 mW m$^{-2}$, respectively, representing a variation of 6.6% in threshold GHF. This calculation highlights the non-negligible role of the ice thickness on $G_{pmp}$ variations and therefore also shows the reduced impact of uncertainties in the GHF data sets on the calculation of the basal temperature.

### 2.4.1 Constraints on GHF

The presence of subglacial water, in the form of lakes or even wet sediments, can inform on the basal temperatures, as it implies the pressure melting point has been reached. This allows for local constraints on models (Pattyn, 2010; Van Liefferinge and Pattyn, 2013). Subglacial lakes are used as in Van Liefferinge and Pattyn (2013) to constrain the GHF data sets. Lakes

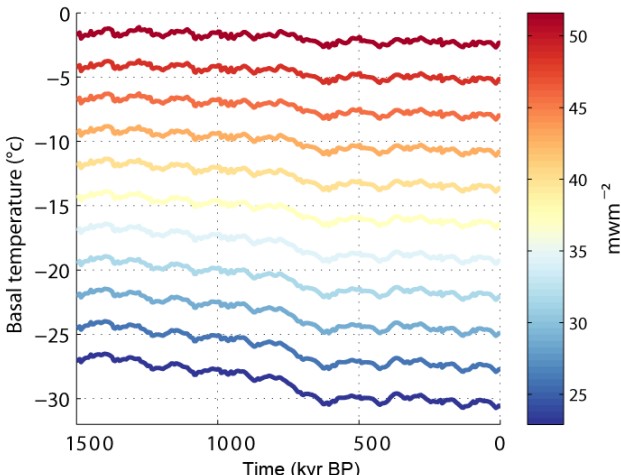

**Figure 2.** Basal temperature evolution at Dome C over a period of 1.5 Ma calculated for an ensemble of GHF values illustrated by colours. Red colours indicate high GHF values which induce temperatures close to the pressure melting point, blue colours show low GHF values which lead to colder basal temperatures. The GHF values on the illustration are restricted between 22 and 52 mW m$^{-2}$.

are considered to be at the pressure melting point, which implies a local GHF value equal or larger than the $G_{pmp}$. In order to calculate the probability of a frozen bed, a Gaussian function is applied to match the GHF data set with the $G_{pmp}$ at lake positions. The value of GHF corresponding to 0.95 ($2\sigma$) of the probability of the cumulative distribution function (CDF) is used at lake locations ($G^c_{pmp}$). On the margin of the influence area, which is 20 km or, if known, the size of the subglacial lake

(particularly relevant for the 54 mapped Dome C survey lakes, Young et al., 2017), a threshold value corresponding to the $G_{pmp}$ is applied. Two cases are possible: the GHF from the data set is higher than the $G_{pmp}$ or the GHF is lower. For both cases, a correction ($G^c$) is made as follows (Pattyn, 2010):

$$G_c(x,y) = G + \left[G^c_{pmp} - G\right] \exp\left[-\frac{x^2 + y^2}{20^2}\right], \tag{7}$$

where $(x, y)$ is the horizontal distance in km from the respective lake positions. Without this correction, subglacial lake areas

would have GHF values corresponding to a frozen bed.

### 2.5 Constraints on Oldest Ice candidate sites

Until now, we have described how to calculate the probability of frozen conditions at the bed. However, the presence of Oldest Ice at depth only allows for a limited range of key ice parameters. Furthermore, we argue that the flatness of the bed should also be taken into consideration as it can affect ice flow and compromise stratigraphic integrity (Dahl-Jensen et al., 2013;

Tison et al., 2015). We introduce this bed topography constraint in the form of the standard deviation of the spatial bedrock topography variability ($\sigma_b$) (Pattyn, 2017; Young et al., 2017; Pollard et al., 2015), which we can assimilate to the roughness

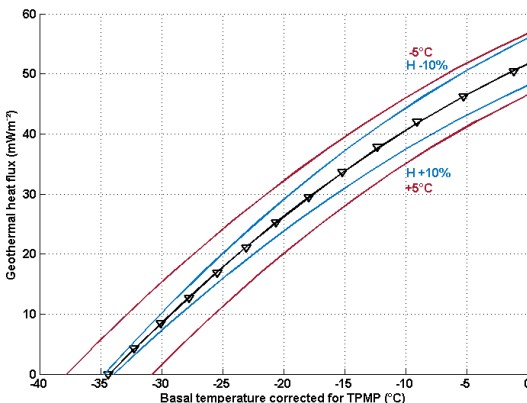

**Figure 3.** Example model result for a location near Dome C. The polynomial fit (black line) indicates the value of GHF needed to keep the bed frozen (corrected for pressure melting). In this example, 13 values were used for the fit calculation (no $H$ or surface temperature uncertainty). The blue lines represent calculated GHF values applying a $H$ uncertainty from the top to the bottom of -10% and +10% respectively and the red lines applying a surface temperature variation of -5°C and + 5 °C from the top to the bottom.

of the bed, calculated over an area of 5 km by 5 km. We also introduce a criterion on the maximum distance from radar data as the density of the radar coverage strongly influences the calculated of the roughness of the bed ($\sigma_b$).

    A previous Antarctic-wide analysis (Van Liefferinge and Pattyn, 2013) used a limit of 2 m a$^{-1}$ for the horizontal surface velocity, an ice thickness larger than 2000 m and cold basal conditions as acceptable ranges for the occurrence of Oldest Ice .

We modify this approach by (i) restricting the parameters' range of values, (ii) taking into account the $G_{pmp}$ instead of the basal temperature, (iii) adding a $\sigma_b$ basal roughness threshold value of 20 m, which implies a relatively smooth bed topography, for Dome Fuji and Dome C areas over a radial distance of 2500 m, (iv) including a threshold of 4 km and 2 km on the maximum distance from radar data (for Dome Fuji and Dome C, respectively). Besides, (v) we use a minimum $H$ value of 2500 m, as we consider that a minimum $H$ value of 2000 m could be inadequate to obtain a sufficient age resolution at the base of the

ice column. We also (vi) use a 1 m a$^{-1}$ threshold for horizontal surface velocities to limit the influence of ice flow. Finally, the choice of a drill site for an Oldest Ice core will have to be within reasonable distance from radar data in order to provide the necessary upstream constraints when reconstructing the ice core's age-depth chronology. This is already taken into account in the constraints listed above.

## 3   Results

### 3.1   Large-scale GHF probability distributions

Threshold $G_{pmp}$ values for the interior of the East Antarctic Ice Sheet for any ice slower than 2 m a$^{-1}$ is displayed in Fig. 4. According to Fig. 4, $G_{pmp}$ varies between 20 and 100 mW m$^{-2}$. Two regions can clearly be distinguished on the map, one with

lower values (in blue), located between South Pole and Dome C, and one with higher values located between the Gamburtsev mountains and Dome Fuji (in red). The difference between both regions is ~10 mW m$^{-2}$. This means that the Dome Fuji area allows for higher values of GHF to keep the bed frozen, compared to, for instance, the Dome C area. The Gamburtsev mountains area differs markedly from other regions, with a high $G_{pmp}$, between 70 and 100 mW m$^{-2}$ (due to thinner ice cover

and lower surface temperatures than at Dome C and Dome Fuji), while the Vostok region presents the lowest $G_{pmp}$ values.

The resulting map of $G_{pmp}$ (Fig. 4) is compared to the published data sets of GHF (Fig. 1; Purucker, 2013; Shapiro and Ritzwoller, 2004; Martos et al., 2017; An et al., 2015). Given that each data set has uncertainties associated to the GHF values, a normal probability density function (PDF) and a normal cumulative distribution function CDF (Fig. 5) can be constructed based on the mean and standard deviation of those values. In our case, the CDF can be interpreted as the probability that $G_{pmp}$

equals or exceeds the GHF of data sets. If the $G_{pmp}$ is lower than the GHF, the probability of having temperate basal conditions is lower.

Our obtained $G_{pmp}$ values are then matched against the CDF (see blue line on Fig. 5) to calculate the probability that ice remained frozen over the last 1.5 Ma. The process is repeated for each of the data sets and the resulting probability is shown in Fig. 6. The An et al. (2015) data set appears to exhibit very low GHF values in comparison with the other data sets, especially

in the dome regions (Fig. 1), which lead to very low probabilities of reaching the pmp at the domes. The probability map is therefore not shown as it is not a major constraint. On a global scale, GHF values are generally higher in the Martos et al. (2017) data set, which results in a overall lower probability of having a frozen bed which is more coherent with the basal temperature map proposed by Pattyn (2010) and Van Liefferinge and Pattyn (2013). Although, the regions with very high or very low $G_{pmp}$ values highlighted in the $G_{pmp}$ distribution map stand out on the three maps, they are most pronounced in

the Shapiro and Ritzwoller data set. The Dome C region is interesting since its values are close to the 0.5 threshold between temperate and freezing conditions. The Dome Fuji region has a higher probability of being below the pressure melting point with probabilities between 0.3 and 0.4 on both Shapiro and Ritzwoller (2004) and Purucker (2013) data set. Regarding Martos et al. (2017) data set, our analysis is more contrasted. The probability of having a non frozen bed is much higher in the north part of Dome Fuji region than in the south.

**3.2   Small-scale GHF probabilities and Oldest Ice: Dome Fuji and Dome C**

The Dome Fuji and Dome C regions are analysed with the same model, but applied at a significantly higher spatial resolution (Fig. 7 and Fig. 8). As expected, results are in line with the previous continental-scale analysis and Dome Fuji generally exhibits higher values for $G_{pmp}$ compared to the Dome C region.

The subglacial highlands 40 km south-west of Dome C, informally named "Little Dome C", and to the north of the Concordia

Subglacial Trench show the lowest probability of being temperate at the base (~0.2), regardless of the presence of subglacial lakes, with a $G_{pmp}$ around 56 mW m$^{-2}$ and 61 mW m$^{-2}$ respectively.

Two sites also emerge in the vicinity of Dome Fuji with $G_{pmp}$ values around 66 mW m$^{-2}$ and a probability to be non-frozen of 0.1. The first site is located to the northwest of the Dome Fuji region (centred on 76°S/30°E, northing-easting 1230/665 km). The other site is located along a topographic feature characterised by a relatively low ice thickness to the southeast of Dome

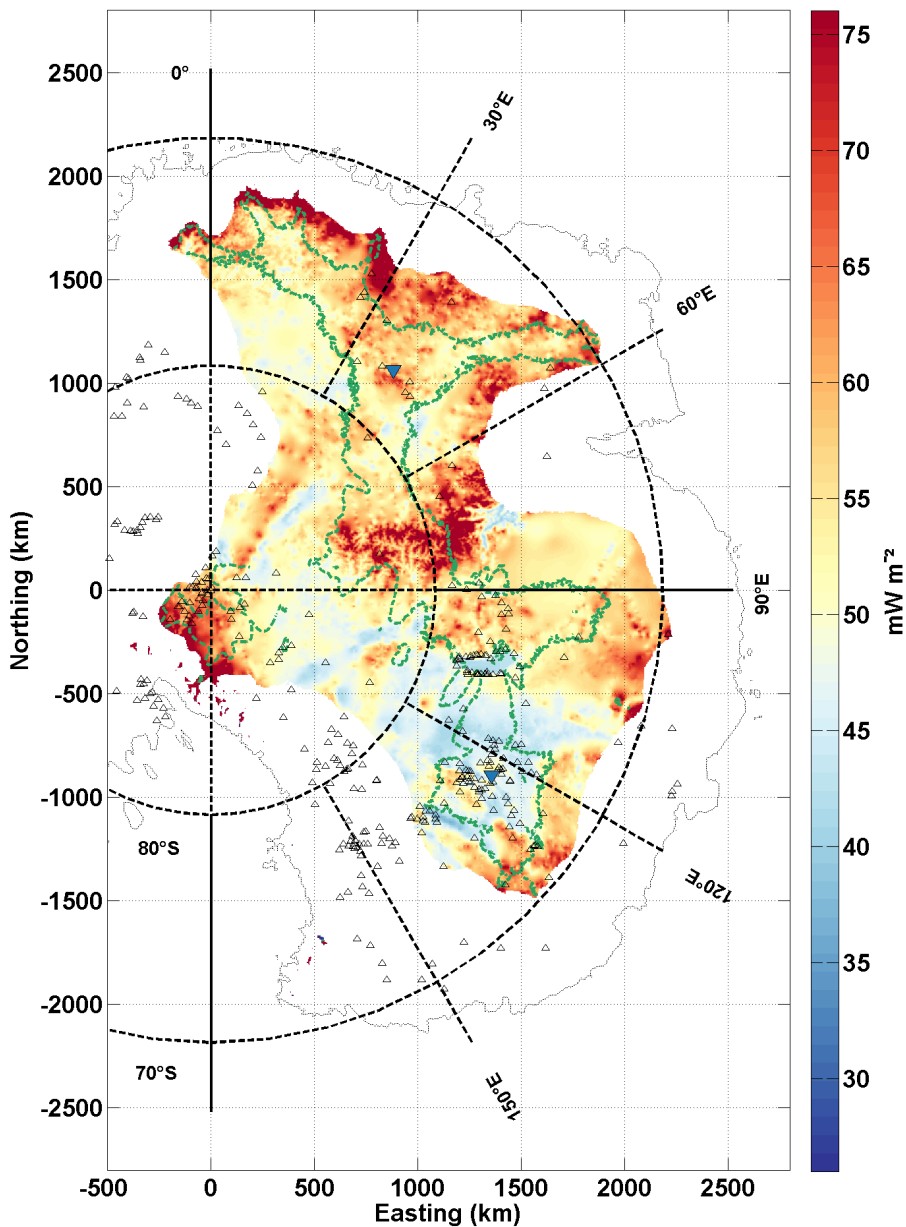

**Figure 4.** Map of $G_{pmp}$, i.e. the maximum GHF to keep a frozen base over 1.5 Ma. Colours represent the magnitude of the GHF (mW m$^{-2}$). The colour scale's central GHF value, in yellow, is 51 mW m$^{-2}$. The small black triangles locate the subglacial lakes (Smith et al., 2009; Wright and Siegert, 2012). The green line outlines areas with surface velocities < 2m a$^{-1}$ (calculated from balance velocities, Pattyn, 2010).

Fuji. As we can see in Fig. 7 and Fig. 8, the Dome Fuji and Dome C sites are well constrained by the new radar surveys recently

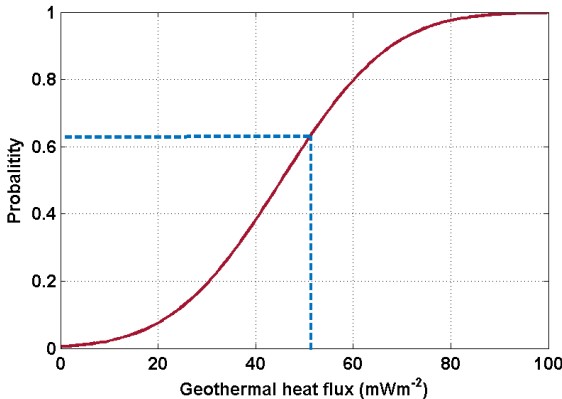

**Figure 5.** The red curve is the cumulative distribution function (CDF) based on the mean and standard deviation of Shapiro and Ritzwoller (2004) GHF data set at Dome C. The blue line is the obtained threshold $G_{\mathrm{pmp}}$ of 51.6 mW m$^{-2}$, and the indicated probability of having a GHF less than that value.

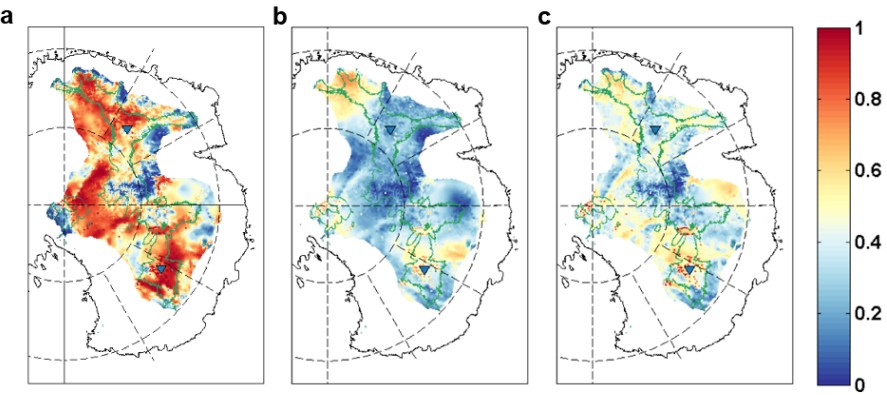

**Figure 6.** Probability that ice reached the pressure melting point over the last 1.5 Ma according to the GHF data sets from Martos et al. (2017), Purucker (2013) and Shapiro and Ritzwoller (2004), from left to right, respectively.

collected (Karlsson et al., 2018; Young et al., 2017), thereby avoiding interpolation errors in ice thickness measurements and in the $G_{\mathrm{pmp}}$ calculation.

The basal topography (Fig. 7 and Fig. 8) shows a smoother bed (lower bed roughness) at Dome Fuji than at Dome C, enhanced by the difference in radar data cover density. In some cases, the steepest slopes are found near bedrock highs (high-lighted by a high $\sigma_b$), and ease with distance from the summit. This is most visible in the vicinity of Dome Fuji (Fig. 7). In the Little Dome C region, the lowest slopes (with a $\sigma_b$ around 15 m) are located towards the edges of the subglacial highlands and in the troughs, also shown by Young et al. (2017). In both regions, basal topography also displays flat-topped mountains. In the

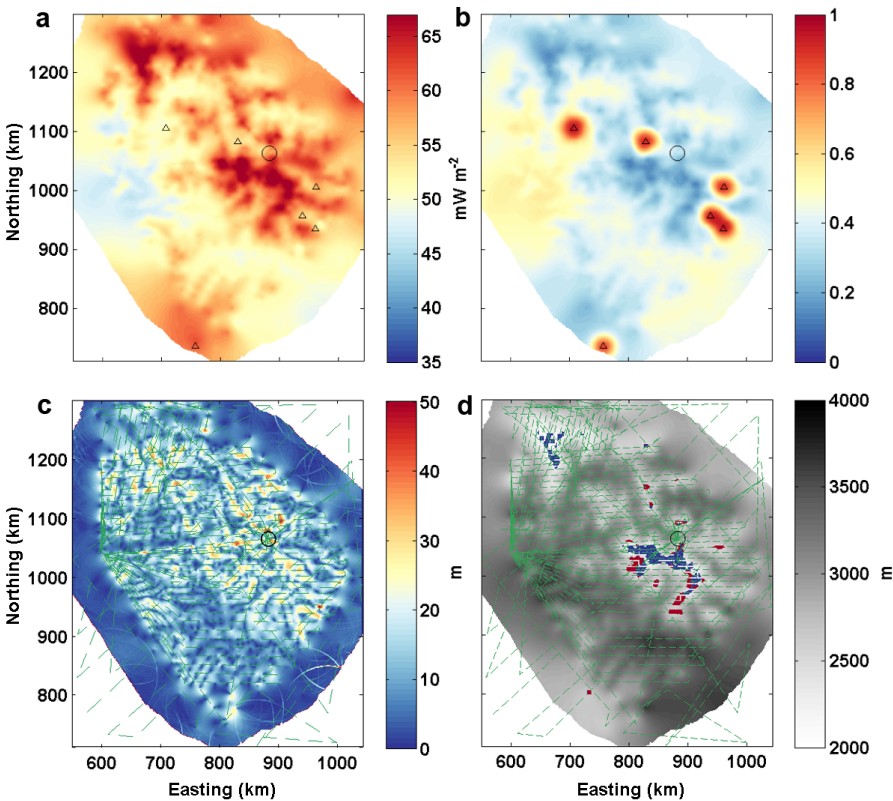

**Figure 7.** A: Map of $G_{pmp}$ results for the Dome Fuji area with the 1D model calculated on a 500 m × 500 m grid, given in a WGS 84 northing-easting coordinate system (km). B: Probability that ice reached the pressure melting point over the last 1.5 Ma according to Shapiro and Ritzwoller (2004). The small black triangles locate subglacial lakes and the circle locates the Dome Fuji ice-core site. C: Standard deviation of bedrock variability; D: Ice thickness from Karlsson et al. (2018). In blue, potential locations of Oldest Ice with $H > 2000$ m, $\sigma_b < 20$m , a probability that ice reached the pressure melting point < 0.4, surface velocity < 2 m a$^{-1}$, distance to radar lines < 4 km. In red, potential locations with $H > 2500$ m and a surface velocity < 1 m a$^{-1}$. The green dashed lines outline the new radar survey (Karlsson et al., 2018).

Dome Fuji region, these plateaus also correspond to high and constant $G_{pmp}$ values as well as lower ice thickness. However, our results are strongly dependent on radar data coverage density as will be explained in section 5.

Regarding Oldest Ice candidate sites at Dome Fuji and Dome C, the red and blue areas on the Fig. 7 D and the Fig. 8 D locate the most promising sites, with in red more conservative parameter values comprising thicker ice and slightly lower ice velocities than in blue. As the ice thickness is mostly larger than 2500 m around Dome C, only red areas are shown. We do not show the same base maps for both data sets because ice thickness has the strongest influence on our model results at Dome Fuji while the bed roughness is more relevant for Dome C due to the difference in spatial radar data resolution (Fig. 7 D and Fig. 8 D). In the vicinity of Dome Fuji, we note three types of promising sites: (1) Extended areas such as those centred on

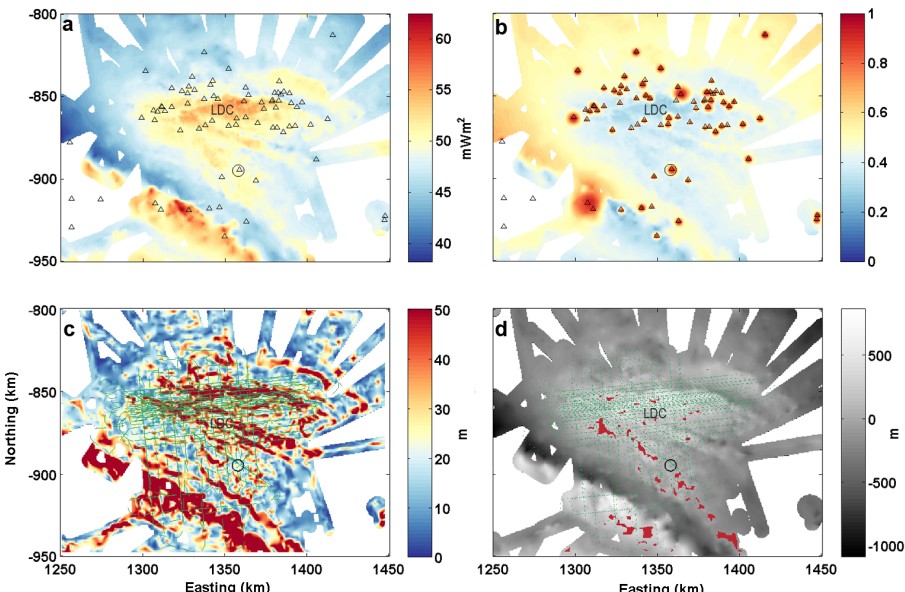

**Figure 8.** A: Map of $G_{\mathrm{pmp}}$ results for Dome C from the 1D model calculated on a 500 m × 500 m grid, given in a WGS 84 northing-easting coordinate system (km). B: Probability that ice reached the pressure melting point over the last 1.5 Ma according to Shapiro and Ritzwoller (2004). The small black triangles locate subglacial lakes (Young et al., 2017) and the circle locates the Dome C ice-core site. C: standard deviation of bedrock variability. D: Basal topography from Young et al. (2017) with in red, potential locations of Oldest Ice for $H > 2500$ m, $\sigma_b < 20$ m, a probability that ice reached the pressure melting point less than 0.4 and distance to radar lines less than 2 km (right). The green dashed lines locate the 2015/2016 radar survey. LDC locates the Little Dome C area as defined by Parrenin et al. (2017) and Cavitte et al. (2018). Fig. 11 is a detailed view of the Little Dome C region.

northing-easting 1210/665 km, 1015/814 km and 1030/875 km, (2) several smaller sites scattered in the vicinity, and (3) areas enclosing domes such as those centred on northing-easting 1090/885 km, 990/848 km or 1150/830. All three types fulfill all conditions. In the vicinity of Dome C, we also highlight a number of promising sites. These are scattered over the Little Dome C subglacial highlands and along a transect from the Dome C ice-core and the northing-easting -860/1315 km (see Fig. 11). In comparison to Dome Fuji, the Dome C sites are less extensive in area, probably due to the denser radar survey coverage.

## 4 Discussion

Knowledge of GHF values at the ice-bedrock interface is a crucial boundary condition for ice flow modelling, yet it remains the most difficult parameter to measure in-situ. Constraining this parameter is therefore essential. From an ice-sheet modelling perspective, it is more realistic to know the temperature gradient at the ice-bed interface rather than a specific GHF value at the interface. The thermal gradient inside the bedrock has an impact on heat availability to the ice (Lowrie, 2007) as does the thermal inertia of the bedrock. Ritz (1987) shows that bedrock temperature will reach equilibrium after thousands of years, on

the scale of several climatic cycles, after a change in ice surface temperature. However knowing the composition, the thickness and the thermal conductivity of the bedrock is also a challenge. At first approximation, we can use a GHF value without taking into account crustal thickness. This simplification is frequently made in glacier and ice sheet models (e.g., Huybrechts et al., 1996; Ritz et al., 1997; Huybrechts and de Wolde, 1999; Pattyn, 2003; Pollard et al., 2005), particularly in steady-state.

At present, the only constraints on basal GHF are provided by remote measurements and modelling approaches. In this work, we quantify the GHF needed to reach the pmp ($G_{pmp}$), and therefore do not calculate an absolute value of the GHF. To do so, we provide constraints on $G_{pmp}$, the threshold value of GHF leading to basal melting, by taking into account the glacial/interglacial history of the ice sheet over 1.5 Ma. Our results generally agree with those of Parrenin et al. (2017) and Passalacqua et al. (2017). However, because of the difference in spatial scales, a more detailed comparison is beyond the

scope of this paper. We will now discuss the influence of the key parameters (surface temperature variations, $\delta\dot{a}$ and $\delta H$) on determining $G_{pmp}$ on locating Oldest Ice candidate sites. For ease of analysis, Fig. 9 summarises their variations. We will demonstrate that the spatial variability of the distribution and the probabilities of $G_{pmp}$ (Fig. 4) are directly related to these parameters.

## 4.1   Surface temperature forcing

Since no high resolution reconstructions of surface temperature currently exist over glacial-interglacial timescales, we have chosen to use present-day surface temperatures (van den Broeke, 2008) generally used in models (Pattyn, 2010; Van Liefferinge and Pattyn, 2013) scaled by the Snyder (2016) surface temperature reconstruction. The Snyder (2016) data set is based on a multi-proxy database and modelling, predicting warmer surface temperatures previous to 800 ka than Lisiecki and Raymo (2005). This global surface temperature data set is controversial as it may overestimate the Earth System Sensitivity to green-

house gases and hence the global-mean surface temperature (Schmidt et al., 2017). In our case, taking into account warmer surface temperatures between 2 Ma and 800 ka represents a conservative boundary condition and therefore increases our confidence in our predictions of Oldest Ice candidate sites. A higher surface temperature will result in a decrease in the advection of cold temperatures towards the base of the ice, therefore decreasing the $G_{pmp}$ and so reducing the probability of finding Oldest Ice. However, as explained in section 3, the higher the value of the GHF, the higher the attenuation of the surface temperature

variations with depth. We have shown that an error in surface temperature has a lesser than, or the same effect as an error in ice thickness. The use of Snyder (2016) as a surface temperature boundary condition should not affect our predictions of Oldest Ice candidate sites as the values lie in the error range (Fig. 3). In addition, absolute differences in temperature from one reconstruction to the next (on the order of a few °C) are dwarfed by differences between a glacial and an interglacial period (on the order of 14°C, Fig. 9). And finally, taking into account all past climate variations (accumulation and ice thickness variation)

reduces the GHF required to reach the pmp and so contributes to a conservative estimate of Oldest Ice candidate sites. The probability maps of frozen bed conditions obtained (Fig. 7 B and Fig. 8 B) refine the Oldest Ice candidate sites first described in Van Liefferinge and Pattyn (2013).

## 4.2 Limits on $G_{\mathrm{pmp}}$ calculation

Surface temperatures and accumulation rates are spatially relatively homogeneous in our regions of interest (Fig. 9 B and D). This is not the case for ice thickness (Fig. 9 C). We can clearly see areas where the mean ice thickness over the last 1.5 Myrs is relatively high, more than 3500 m (Dome C, Vostok) and other areas where the mean ice thickness is lower (Gamburtsev mountains) with differences over time on the order of 1000 m. This is also the case for the Lakes District and Dome C areas where ice thickness variations are large, around 200 m. In our model this thickness variation corresponds to a $G_{\mathrm{pmp}}$ decrease or an increase of 10 mW m$^{-2}$.

The variations between the higher and the lower accumulation are around 0.03 m a$^{-1}$ for the whole period. In a very extreme scenario, we can also consider this fluctuation as the maximum error of our simulation. This gives us a $G_{\mathrm{pmp}}$ variation on the order of +5.5 and -6.4 mW m$^{-2}$, for an increase and a decrease in accumulation, respectively. For changes in surface temperature, the difference in $G_{\mathrm{pmp}}$ is then +7.6 and -7.2 mW m$^{-2}$. The combination of the two errors indicates a variation of +13.1 and -12.7 mW m$^{-2}$, respectively. Whereas $H$ dominates our result for the $G_{\mathrm{pmp}}$ calculation, errors in accumulation and surface temperature can also have a major impact on the $G_{\mathrm{pmp}}$, on the order of 25% of the $G_{\mathrm{pmp}}$ value.

Dome Fuji and Dome C are interesting locations to look at in detail as they provide direct measurements of the temperature profile from ice core measurements. It is therefore possible to deduce the GHF at the base under present conditions. Our $G_{\mathrm{pmp}}$ value at Dome Fuji is 57.3 mW m$^{-2}$, which agrees with values previously calculated by Seddik et al. (2011) and Hondoh et al. (2002) of 60 and 59 mW m$^{-2}$, respectively, by taking into account a small amount of basal melting. The GHF calculated from the temperature gradient from the Dome Fuji deep ice core is 51.5 mW m$^{-2}$. In comparison, the four GHF data sets show relatively low values (between 40 and 59 mW m$^{-2}$) except the value of 65 mW m$^{-2}$ provided by Martos et al. (2017). In the Dome Fuji region, the bed has a high probability of being close to the pmp , according to the four data sets, but the calculated value of the basal temperature gradient at Dome Fuji points to conditions that are likely non-frozen. This comparison increases the confidence in our approach, as does the comparison with the steady-state approach, described in the following section.

## 4.3 Steady-state model comparison

The spatial variability of GHF noted in the $G_{\mathrm{pmp}}$ maps derived above is also visible in the results of the Van Liefferinge and Pattyn (2013) simple model and the new recalculation of Karlsson et al. (2018). In both models, we observe a clear spatial variability of the GHF, which mainly reflects the ice thickness of the ice sheet. In the GHF histogram calculated for both models (transient and steady state), we clearly note a difference in the mode of the distributions. The difference is  5 mW m$^{-2}$, with lower values for the steady-state model. This is also what emerges from Fig. 10 that shows differences in GHF between the Van Liefferinge and Pattyn (2013) simple model and the $G_{\mathrm{pmp}}$ calculated. The major part of the highlighted region (Fig. 10) shows GHF values corresponding to the pressure melting point which are $\sim$5 mW m$^{-2}$ lower for the steady state model except in the area between Dome C and South Pole, where the difference is in some places sightly positive or close to zero, explained by a lower $\dot{a}$ and $\delta H$. We attribute the lower values in our previous steady-state model (simple model) to failure in taking into account variations in surface temperature coupled with changes in thickness. A steady state model will produce

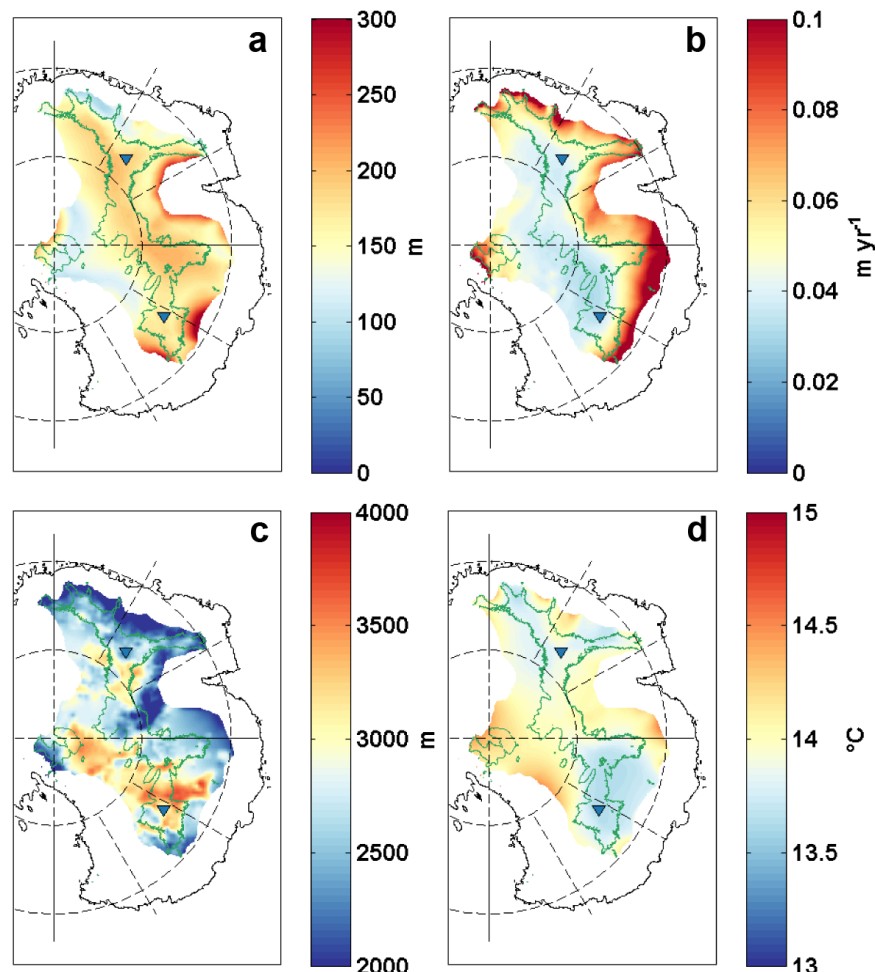

**Figure 9.** Paleo-reconstructions for the Antarctic Ice Sheet over the last 1.5 Ma. A: Ice thickness variation from Pollard and DeConto (2009). The colour scale is truncated at 300 m. B: Surface mass balance changes (m a$^{-1}$) related to surface temperature variations (Pollard and DeConto, 2012). C: Mean ice thickness (m) from Pollard and DeConto (2009). D: Reconstructed variation in the amplitude of surface temperature (°C) forced by the multi-proxy database and modelling of Snyder (2016).

an amplitude of basal temperature variations similar to the surface temperature variations, but a transient model leads to a much smaller amplitude, on the order of 3°C at the base for a surface variation of 14°C. If ice thickness is reduced, advection of cold temperatures at the surface is increased, which in the transient model implies a decreased basal temperature and a higher value of $G_{pmp}$.

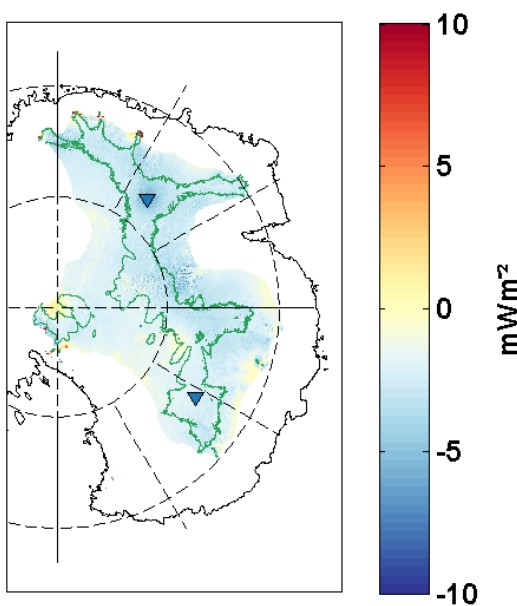

**Figure 10.** Comparison of the GHF needed to reach the pressure melting point calculated using the Van Liefferinge and Pattyn (2013) simple model and the $G_{pmp}$ calculated in this paper (mW m$^{-2}$). Blue colours (negative values) indicate that we need a higher GHF for the $G_{pmp}$ to reach the pressure melting point, and vice versa for the positive values (in red).

## 5  Conclusion and implications for Oldest Ice candidate sites

Although there is a large number of parameters that can influence the presence/absence of Oldest Ice at depth, our modelling approach identifies and constrains the key parameters. The obtained $G_{pmp}$ probability maps have a strong dependency on the spatial resolution of the input data sets, namely the horizontal resolution of the GHF data sets and the horizontal spacing of the radar surveys used. Additionally, a direct comparison of Dome Fuji and Dome C is precluded by the difference in spatial resolution of their respective radar data sets. We note that the bed roughness ($\sigma_b$) is lowest in regions where radar line spacing is the widest, clearly visible on the marginal regions of the $\sigma_b$ maps (Fig. 7 C and Fig. 8 C). To take into account the influence of the resolution of the radar surveys, we restrict ourselves to a maximum radial distance from any radar line of 4 km for the Dome Fuji region and 2 km for the Dome C region, so as to take into account the horizontal resolution of their respective radar surveys and reduce the uncertainty in bed topography roughness.

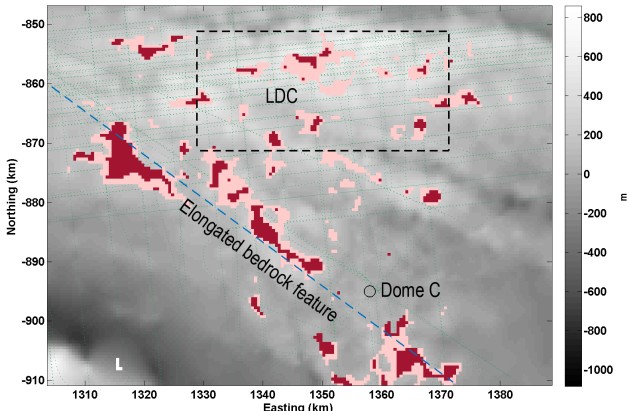

**Figure 11.** Detail of promising sites around Little Dome C. Potential locations of Oldest Ice are shown in red for $H > 2500$ m, $\sigma_b < 20$ m, a probability that ice has reached the pressure melting point less than 0.4 and distance to radar lines less than 2 km. In pink, potential locations of Oldest Ice with the same parameter values but a value of $\sigma_b < 30$ m. The background displays basal topography from Young et al. (2017),. The blue dashed line locates the "elongated bedrock feature" discussed in the paper. The green dashed lines locate the 2015/2016 radar survey. LDC and the dashed rectangle locate the Little Dome C area as defined by Parrenin et al. (2017) and Cavitte et al. (2018).

### 5.1 Dome Fuji

Dome Fuji shows lower $\sigma_b$ values on average due to the low density of the radar coverage. The region shows high $G_{\mathrm{pmp}}$ values combined with a thin ice cover. Therefore, the spatial variations in ice thickness, $H$, dominate the distribution of Oldest Ice for this region. The most promising Oldest Ice candidate sites in the vicinity of Dome Fuji are located on the edges of subglacial

mountains, which have the advantage of offering a thicker $H$, a lower $\sigma_b$ while keeping cold conditions at the base. Plateau areas also show potential Oldest Ice sites, but these are less promising due to their lower age resolution as a result of their thinner ice cover.

### 5.2 Dome C

Dome C is characterised by higher values of $\sigma_b$ on average due to the radar coverage's higher spatial density. We note that our

$\sigma_b$ distribution is similar to that calculated by Young et al. (2017), which adds confidence in our results. The bed roughness and the $G_{pmp}$ probability distributions have the strongest influence on the location of candidate sites for this region. In the vicinity of Dome C, potential candidate sites are found in two areas in particular (Fig. 11): near Little Dome C where $\sigma_b$ values are low and the probability of a frozen bed is high, as well as along a transect from the Dome C ice-core and the northing-easting -860/1315. The geometry of this elongated bedrock feature is evocative of a raised fault block (also referred to as a horst in

geology) which, if confirmed, implies an uplifted but relatively flat bedrock surface. This is promising because it offers a wider area with appropriate ice thicknesses for the recovery of Oldest Ice.

We conclude that, following the analysis of the recent radar data surveys and our modelling efforts, both regions remain interesting as Oldest Ice drilling sites. This work highlights a number of candidate locations that will benefit from the collection of additional geophysical data and modelling.

*Competing interests.* The authors declare that they have no conflict of interest.

*Acknowledgements.* We would like to first thank the Editor Eric Larour and both reviewers for their constructive reviews on the paper. This publication was generated in the frame of Beyond EPICA-Oldest Ice (BE-OI). The project has received funding from the European Union's Horizon 2020 research and innovation programme under grant agreement No. 730258 (BE-OI CSA). It has received funding from the Swiss State Secretariate for Education, Research and Innovation (SERI) under contract number 16.0144. It is furthermore supported by national partners and funding agencies in Belgium, Denmark, France, Germany, Italy, Norway, Sweden, Switzerland, The Netherlands and the United

Kingdom. Logistic support is mainly provided by AWI, BAS, ENEA and IPEV. The radar survey at Dome C was supported by NSF grant PLR-1443690, and ADD for logistic. B. Van Liefferinge received a "Fonds Van Buuren" grant to complete this research. Computational resources have been provided by the Shared ICT Services Centre, Université libre de Bruxelles. We would like to thank David Pollard for the data and the fruitful discussions.

The opinions expressed and arguments employed herein do not necessarily reflect the official views of the European Union funding agency,

the Swiss Government or other national funding bodies.

This is UTIG contribution #3261.

This is BE-OI publication number #6 .

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
