# Peer review of "Promising Oldest Ice sites in East Antarctica based on thermodynamical modelling"

_The Cryosphere, 2017_

## Referee Comment (RC1) · R. Greve (Referee) · 26 May 2018

This study deals with the hot topic of detecting a suitable site for an "Oldest Ice" core in Antarctica. It is essentially an update of an earlier study by Van Liefferinge and Pattyn (2013). The authors use a transient 1D model, forced by time-dependent atmospheric conditions and ice thickness over the last 2 Ma, and compute the basal temperature as a function of the geothermal heat flux (GHF). By comparing with existing GHF data, the authors compute probability maps for the basal temperature having reached the pressure melting point. Combination with other criteria (flow speed, basal roughness etc.) allows identifying several candidates for Oldest Ice sites in the vicinity of Dome C

and Dome F.

The findings are certainly not the final word on the matter, but the paper constitutes a significant progress, and I am generally in favour of publication. However, I would like to raise some points the authors should consider:

Throughout MS: "Puruker" -> "Purucker"

P2, L3: Add a semicolon after "ice sheet".

P2, L24: "of the crustal" -> "of the crust"

P2, L25/26, "it is crucial to know basal temperature gradients at the ice-bedrock interface and not GHF within the crust": This is not necessarily true and depends on the type of the modelling study. For longer-term (e.g., glacial-interglacial cycle) modelling studies, it is more physical to use the GHF within the crust and apply it as a boundary condition some kilometres below the ice-bedrock interface.

P4, Fig. 1: If I'm not misled, this figure is not referenced anywhere in the text.

P5, L17/18, "horizontal advection may safely be neglected": What about horizontal conduction?

P5, Eq. (2): This boundary condition only holds for a cold base.

P5, Eq. (3): This equation should be given a reference.

P5, Eq. (3) and L26 vs. P6, L3 and Eq. (5): The surface accumulation rate should consistently be denoted by either "a" or "$\dot{a}$".

P6, Table 1: Why not using the more realistic, temperature-dependent representations of the thermal conductivity and the heat capacity? For the large range of temperatures relevant for Antarctica, the dependence is significant (e.g., Greve and Blatter 2009, "Dynamics of Ice Sheets and Glaciers").

P6, L10: I think it is problematic to keep the bed elevation constant in time and then apply a time-varying ice thickness produced by a model that includes isostatic adjustment (Pollard and DeConto 2009). This procedure overestimates surface elevation variability and thus surface temperature variability over time. Why not including a simple local-lithosphere-relaxing-asthenosphere model? This should be easy to implement, not consume much extra computing time, and it is quite realistic for the interior of Antarctica due to the enormous horizontal extent.

P7, L2: "500 m by 500" -> "500 by 500 m"

P10, L3/4, "Although, the regions highlighted...": I don't understand this sentence.

P10, L7: "our analyse is more contrasted" -> "our analysis is more contrasted"

P11, L4/5: I'm not sure whether the Shapiro and Ritzwoller (2004) GHF values are the best reference. If I interpret Figs. 7b and 8b correctly, this produces probabilities of Dome F and Dome C having reached the PMP of $\sim$0.3 and 0.5, respectively. However, if I remember well, direct observations have shown that both ice cores are warm-based today. This challenges the credibility of the presented results. Further, I have found that the Martos et al. (2017) data generally produce better results for ice flow and basal temperature in 3D, large-scale simulations of Antarctica (recent work, unpublished).

P12, L4: "on Fig. 7" -> "in Fig. 7"

P14, L22: "the values lies" -> "the values lie"

P14, L26: I think the reference to Fig. 9 is wrong.

P14, L29, "Spatial and temporal forcing variations with respect to surface temperature and accumulation rate are relatively limited (Fig. 9)": Looking at Fig. 9, these variations don't seem to be so small.

P15, L15, "high probablity of being below or close to the pmp": What is meant by "below or close"?

P16, L8, "maximum radial distance of 4 km and 2 km from Dome Fuji and Dome C":

Are these numbers correct? If so, I don't understand it. Earlier in the paper (Figs. 7 and 8), much larger windows around these two sites were discussed. How does this go together?

---

## Referee Comment (RC2) · Anonymous Referee #2 · 2 Jul 2018

The study by Van Liefferinge et al titled "Promising Oldest Ice sites in East Antarctica based on thermodynamical modelling" deals with an important topic in this field. It is timely and worthwhile publishing after revising several points that need clarification as following.

Comments and questions

(1) The surface temperature, accumulation, ice thickness and GHF are important boundary conditions, but the surface temperature data used in this present study for both Dome F and Dome C are not clear. Please clarify them (in a table, for example), and discuss the uncertainty and its influence upon the results in the discussion section.

[Figure]

(2) The vertical flow that is used in the advection term is based on the equation (2), which is based on Pattyn, 2010 and Van Liefferinge and Pattyn, 2013, but this is certainly an approximation. Please discuss the effect of the assumption that is made on the result in the discussion section. For example, Parrenin et al, 2017 use different assumption but how does the difference affect the present work?

(3) Neglecting the rock temperature calculation would overestimate the amplitude between the glacial-interglacial condition and then can underestimate the GHF limit for the oldest ice. Please discuss this problem in the discussion section.

(4) Why is the threshold of ice thickness to find the oldest ice for DomeF and Dome C different, 2000m and 2500m, respectively?

(5) How was the lower limit of ice thickness to find the oldest ice determined?

(6) For Dome C, Parrenin et al, 2017 show the location of melt and estimate the possible GHF from modeling and radar data analysis. Discuss what we learn from the present study after knowing the publication of Parrenin et al 2017.

(7) Plot the "Little Dome" and Dome C in the map (or show both locations) and discuss the result related to this area.

Some minor comments:

P.3 L10-12: refer to the data of GHF known at those cites.

P.3 L17-18, I cannot understand what you mean by this sentence, "Furthermore, the mechanisms that control the geometry. . . . . . ".

P.4 Figure1: please show the temperature change of Dome F, too.

P.6 Please show the map of Ts obs (the observed surface temperature) including Dome C and Dome F. It is not clear which dataset is used.

P.6 Ice thickness history taken from Pollard and Deconte, 2009 should be shown (perhaps in Figure 1) for the aid of understanding the results.

P.8 L12, "adding …...basal roughness threshold value of 20m….". The meaning is not clear.

P.9 Figure3. Is this for Dome C? Please show both Dome C and Dome F and explain the difference.

P.9 L8, "mainly due to shallower ice": how much caused by the ice thickness and surface temperature?

P. 10 Figure 4. Show in the caption that the area with ice flow within 2 m/yr is displayed.

P. 11 L2-3, very good that the higher spatial resolution is shown and discussed in the following section, but the Figure 7 and Figure 8 should be displayed in the same resolution.

P.11, L7, show the latitude and longitude of "Little Dome C".

P. 12 Why is the red area around the triangle larger in Fig.7 (b) than Fig. 8 (b)? I cannot understand how this was determined.

P.12 Figure 7. (d) displays the ice thickness but the Figure 8. (d) displays the Basal topography. For the readers' understanding, it is better to use the common variable (either ice thickness or basal topography)

P.12 L7 "lower bed roughness at Dome Fuji than Dome C" is not easy to understand.

P.13 Figure8 (a) and (b): Why are the number of triangles and their location different in the two figures?

P. 16 Figure 9 (d) Why does Snyder (2016) show the "map" of surface temperature change? Snyder 2016 is only providing a time series.

P. 16 Figure 9 (d): Surface temperature changes between 1.5Ma and 0 Ma? This is not possible. Please check and rewrite what you mean.

P.17 Figure 10: Clarify which difference (and the sign) is meant.

P.17 L6, explain more why " less promising due to thinner ice cover". Thinner ice could be promising in freezing condition. Discuss the advantage and disadvantage.

P.17 L11: Where are the "two areas"?

P.17 L13: "evocative of a horst" is not understandable.

P.18 L4: "a number of candidate locations" are not clear in the figures. Please make a summary figure to enlarge and focus the locations.

---

## Author Comment (AC1) · 18 Jul 2018

**Response to reviewer comments RC1 on Promising Oldest Ice sites in East Antarctica based on thermodynamical modelling**

We would like to first thank the Editor Eric Larour and both reviewers for their constructive reviews on the paper. In order to address all comments, you will find here our answers point-by-point, which sometimes creates some repetitive answers. We hope that we have satisfactorily responded to all comments and remarks, which can be found here below.

We would also like to bring to your attention that:

(1) We have updated the references: Cavitte et al., 2018 which is now published, and Karlsson et al., 2018 which is now accepted for publication (in press)

(2) A few of the figures have been increased in size for clarity, in addition to the relevant changes for the reviews

(3) We have made a few minor additional wording and aesthetic changes throughout the manuscript.

Reviewer comments are in black, answers in blue and text edits in red.

Throughout MS: "Puruker" -> "Purucker". Changed

P2, L3: Add a semicolon after "ice sheet". Added

P2, L24: "of the crustal" -> "of the crust". Changed

P2, L25/26, "it is crucial to know basal temperature gradients at the ice-bedrock interface and not GHF within the crust": This is not necessarily true and depends on the type of the modelling study. For longer-term (e.g., glacial-interglacial cycle) modelling studies, it is more physical to use the GHF within the crust and apply it as a boundary condition some kilometres below the ice-bedrock interface.

This is indeed an important point. We copy here our response to RC2: Most methods calculate average GHF values within or at the surface of the crust, without accounting for gradients of GHF within the crust. From an ice-sheet modelling perspective, it is more realistic to know the temperature gradient at the ice-bed interface rather than a specific GHF value at the interface. The thermal gradient inside the bedrock has an impact on heat availability to the ice (Lowrie, 2007) as does the thermal inertia of the bedrock. Ritz (1987) shows that bedrock temperature will reach equilibrium after thousands of years, on the scale of several climatic cycles, after a change in ice surface temperature. She shows that the use of a 2 km thick crust for the calculation of the crustal thermal gradient is enough to accurately model the changes induced by surface temperature variations. For climate cycles with a 100 kyr cyclicity, the basal temperature perturbation at the bed is ~40% of the surface temperature perturbation if crustal thickness isn't taken into account, and 15% if it is. However knowing the composition, the thickness and the thermal conductivity of the bedrock is also a challenge. At first approximation, we can use a GHF value without taking into account crustal thickness. This simplification is frequently made in glacier and ice sheet models (e.g. Huybrechts et al., 1996; Ritz et al., 1997; Huybrechts and de Wolde, 1999; Pattyn, 2003; Pollard et al., 2005), particularly in steady-state.

We have modified our discussion to describe this simplification explicitly. The paragraph "Knowledge of GHF values at the ice-bedrock interface is a crucial boundary condition for ice flow modelling, yet it remains the most difficult parameter to measure in-situ. Constraining this parameter is therefore essential. GHF is determined by the geology of the bedrock (type of rock, presence of volcanism, etc). However, bedrock geology is unknown in the Antarctic interior and therefore cannot be taken into account in our model" is changed to "From an ice-sheet modelling perspective, it is more realistic to

know the temperature gradient at the ice-bed interface rather than a specific GHF value at the interface. The thermal gradient inside the bedrock has an impact on heat availability to the ice (Lowrie, 2007) as does the thermal inertia of the bedrock. Ritz (1987) shows that bedrock temperature will reach equilibrium after thousands of years, on the scale of several climatic cycles, after a change in ice surface temperature. However knowing the composition, the thickness and the thermal conductivity of the bedrock is also a challenge. At first approximation, we can use a GHF value without taking into account crustal thickness. This simplification is frequently made in glacier and ice sheet models (e.g. Huybrechts et al., 1996; Ritz et al., 1997; Huybrechts and de Wolde, 1999; Pattyn, 2003; Pollard et al., 2005), particularly in steady-state."

P4, Fig. 1: If I'm not misled, this figure is not referenced anywhere in the text. Fig. 1 was mentioned in the Results section but we have now referenced this figure earlier in the manuscript, in section 2.3 first, and several times later on.  Finally, background temperature changes $\Delta T(t)$ are taken from the reconstruction of Snyder (2016), discussed in the section 4.1, scaled to Dome C ice-core temperature reconstruction (Parrenin et al., 2007) (Fig. 1).

P5, L17/18, "horizontal advection may safely be neglected": What about horizontal conduction?
This is a good point. Ignoring horizontal heat conduction is widely used as a 1D approximation as well.

Horizontal heat conduction (horizontal diffusivity) may have an effect if the ice thickness is changing rapidly over short distances. For relatively constant ice thickness within windows of the order of magnitude of an ice thickness (3-4 km), the conduction will be low as horizontal temperature gradients and second derivatives are small (because surface temperatures are rather constant over such distances. This is not the case in the vertical, where both T gradients (order of 50K over 3km) and second gradients (shape of the T profile is not linear) are large.
We changed the sentence as follows: In divide-adjacent areas, horizontal advection and horizontal heat conduction may safely be neglected as for areas with a relatively smooth bed, horizontal conduction is much lower than vertical conduction (Hindmarsh, 1999, 2018).

P5, Eq. (2): This boundary condition only holds for a cold base. We agree and therefore explicitly state this in the sentence as follows:
The basal boundary condition for a cold base bed is given by

P5, Eq. (3): This equation should be given a reference. Added. (Hindmash, 1999; Pattyn, 2010)

P5, Eq. (3) and L26 vs. P6, L3 and Eq. (5): The surface accumulation rate should consistently be denoted by either "a" or "ndot{a}". We agree, this was a mistake as we always use accumulation as rate. We have therefore changed the notation everywhere to "ndot{a}".

P6, Table 1: Why not using the more realistic, temperature-dependent representations of the thermal conductivity and the heat capacity? For the large range of temperatures relevant for Antarctica, the dependence is significant (e.g., Greve and Blatter 2009, "Dynamics of Ice Sheets and Glaciers"). The reviewer makes a good point. We made the test following e.g. Ritz (1987) given $K_i = 9.828 \exp (-0.0057 T)$ in W m$^{-1}$ K$^{-1}$, where $K_i$ is the conductivity depending on the temperature (T) inside the ice.
Using a temperature-dependent representation of parameters has very little impact in our calculations. We did a test close to Dome C site with a Gpmp calculated with fixed parameters at 51.7 mW m$^{-2}$. The same calculation with temperature-dependant parameters gives a value of 53.6 for the Gpmp (3.5 % of difference). Taking temperature-dependant parameter constrains our result less as the Gpmp is higher.

P6, L10: I think it is problematic to keep the bed elevation constant in time and then apply a time-varying ice thickness produced by a model that includes isostatic adjustment (Pollard and DeConto

2009). This procedure overestimates surface elevation variability and thus surface temperature variability over time. Why not including a simple local lithosphere-relaxing-asthenosphere model? This should be easy to implement, not consume much extra computing time, and it is quite realistic for the interior of Antarctica due to the enormous horizontal extent. This is definitely an important remark, and probably due to an unclear description of our method. Strictly speaking the variation in surface temperatures shown in the manuscript is obtained by scaling present day temperatures with paleo ice elevation (given by Pollard and DeConto, 2009). We do not use present day bed elevations for the model results but only use ice thickness variations given by Pollard and DeConto (2009). To summarise, we didn't use a bed relaxation model but we used the bed elevation variations and ice thickness variations given by Pollard and Deconto (2009) that already take isostatic adjustment into account. We have now changed the paragraph to: Surface elevation changes with time are obtained from changes in ice thickness with time obtained from a model that takes into account isostatic adjustment, given by s(t) = b+H(t), where b is the varying bed elevation varying with time and H(t) the time-varying ice thickness, …

P7, L2: "500 m by 500" -> "500 by 500 m". Changed

P10, L3/4, "Although, the regions highlighted...": I don't understand this sentence.
    We wanted to refer to regions with very high or very low $G_{pmp}$ values. The sentence was therefore changed to:
    Although, the regions with very high or very low $G_{pmp}$ values highlighted in the Gpmp distribution map stand out on the three maps…

P10, L7: "our analyse is more contrasted" -> "our analysis is more contrasted". Changed

P11, L4/5: I'm not sure whether the Shapiro and Ritzwoller (2004) GHF values are the best reference. If I interpret Figs. 7b and 8b correctly, this produces probabilities of Dome F and Dome C having reached the PMP of _0.3 and 0.5, respectively. However, if I remember well, direct observations have shown that both ice cores are warm-based today. This challenges the credibility of the presented results. Further, I have found that the Martos et al. (2017) data generally produce better results for ice flow and basal temperature in 3D, large-scale simulations of Antarctica (recent work, unpublished). We agree this was unclear. We removed the confusing and not correct sentence "" Martos et al. (2017) use a completely different method than Shapiro and Ritzwoller (2004). In general, Shapiro and Ritzwoller (2004) obtain higher values of GHF in the interior of the ice sheet compared to Purucker (2013) and An et al. (2015), but lower than Martos et al. (2017). However, looking at dome areas, Martos (2017) GHF values are clearly higher than any of the other methods' results. Here below, we show a table of GHF values at the domes for the five different published data sets. Martos et al. (2017) data does have the advantage of a higher spatial resolution which could be beneficial for 3D model calculations, but a higher resolution can result simply from model mesh refinement and not GHF knowledge accuracy. Fig.6 shows the probability for 3 data sets: xx, xx and Martos et al. (2017). We can clearly see that Martos et al. (2017) show the highest probability of being at the pmp in the dome regions (see Fig. below). Instead of choosing what is the "best GHF data set", we have opted here to use all published data sets together so that they balance out their strengths and weaknesses.

[Figure]

*Probability that ice reached the pressure melting point over the last 1.5 Ma according to: a) Purucker (2013) b) Martos et al. (2017) c) An et al. (2015) d) Shapiro and Ritzwoller (2004)*

So model results shown in Fig 7d and 8d use all five GHF datasets to constrain the promising sites. However, in panels b of each of those figures, we simply chose to display one of the five GHF datasets. In this case, we chose Shapiro and Ritzwoller (2004) as their mean GHF is less extreme than others.

*Geothermal heat flux values at the domes in mW m-2*

|  | Dome Fuji | Dome A | Dome C |
|---|---|---|---|
| Shapiro and Ritzwoller (2004) | 50 | 47 | 45 |
| Fox Maule et al. (2005) | 59 | 53 | 56 |
| Puruker (2013) | 40 | 36 | 42 |
| An et al. (2015) | 40 | 46 | 44 |
| Martos et al. (2017) | 65 | 54 | 58 |
| Median value | 50 | 47 | 45 |
| Standard deviation | 11 | 7 | 7 |

P12, L4: "on Fig. 7" -> "in Fig. 7". Changed

P14, L22: "the values lies" -> "the values lie". Changed

P14, L26: I think the reference to Fig. 9 is wrong.
Apologies, this sentence was corrected.
The probability maps of frozen bed conditions obtained (Fig. 7 B and Fig. 8 B) refine the Oldest Ice candidate sites first described in Van Liefferinge and Pattyn (2013).

P14, L29, "Spatial and temporal forcing variations with respect to surface temperature and accumulation rate are relatively limited (Fig. 9)": Looking at Fig. 9, these variations don't seem to be so small. We agree that variations can be larger for these variations. But what we meant here is that, if look at areas of interest for Oldest Ice (i.e. where velocity is less 2 m/year), surface temperature and surface accumulation rates don't vary much from one dome to the next, while mean ice thickness and variations in ice thickness are spatially heterogeneous for our areas of interest. We have changed the paragraph to:

Surface temperatures and accumulation rates are spatially relatively homogeneous in our regions of interest (Fig. 9 B and D).

P15, L15, "high probability of being below or close to the pmp": What is meant by "below or close"?
We agree this statement is a little useless since temperature of the bed cannot be above the pmp. We have therefore changed the sentence as follows:
The bed in the Dome Fuji region has a high probability of being close to the pmp.

P16, L8, "maximum radial distance of 4 km and 2 km from Dome Fuji and Dome C": Are these numbers correct? If so, I don't understand it. Earlier in the paper (Figs. 7 and 8), much larger windows around these two sites were discussed. How does this go together?
Our sentence was confusing. In both cases the criteria is the same: we require a distance to radar lines less than 2 km or 4 km for Dome Fuji and Dome C, respectively. And not only _from_ Dome F or Dome C (the "from" was problematic). We changed the sentence to:
To take into account the influence of the resolution of the radar surveys, we restrict ourselves to a maximum radial distance from any radar line of 4 km for the Dome Fuji region and 2 km for the Dome C region.

---

## Author Comment (AC2) · 18 Jul 2018

**Response to reviewer comments RC2 on Promising Oldest Ice sites in East Antarctica based on thermodynamical modelling**

We would like to first thank the Editor Eric Larour and both reviewers for their constructive reviews on the paper. In order to address all comments, you will find here our answers point-by-point, which sometimes creates some repetitive answers. We hope that we have satisfactorily responded to all comments and remarks, which can be found here below.
We would also like to bring to your attention that:

(1) We have updated the references: Cavitte et al., 2018 which is now published, and Karlsson et al., 2018 which is now accepted for publication (in press).
(2) A few of the figures have been increased in size for clarity, in addition to the relevant changes for the reviews.
(3) We have made a few minor additional wording and aesthetic changes throughout the manuscript.

Reviewer comments are in black, answers in blue and text edits in red.

Comments and questions

(1) The surface temperature, accumulation, ice thickness and GHF are important boundary conditions, but the surface temperature data used in this present study for both Dome F and Dome C are not clear. Please clarify them (in a table, for example), and discuss the uncertainty and its influence upon the results in the discussion section.
We noticed that it was not clear where the different datasets came from. We now clearly state where the data comes from. In consequence we changed paragraph 2.3 (Model forcing):
Atmospheric forcing is applied in a parameterized way, based on the observed fields of surface mass balance (accumulation rate) based on the output of the RACMO regional atmospheric climate model over the period 1980-2004, calibrated with observed surface mass balance rates (van de Berg et al., 2006; van den Broeke et al., 2006) and surface temperature (van den Broeke, 2008).
And paragraph 4.1 (Surface temperature forcing):
Since no high resolution reconstructions of surface temperature currently exist over glacial-interglacial timescales, we have chosen to use present-day surface temperatures (van den Broeke, 2008) generally used in models (Pattyn, 2010; Van Liefferinge and Pattyn, 2013) and scaled by the Snyder (2016) surface temperature reconstruction. The Snyder (2016) data set is based on a multi-proxy database and modelling, predicting warmer surface temperatures previous to 800 ka than Lisiecki and Raymo (2005). This global surface temperature data set is controversial as it may overestimate the Earth System Sensitivity to greenhouse gases and hence the global-mean surface temperature (Schmidt et al., 2017). In our case, taking into account warmer surface temperatures between 2 Ma and 800 ka represents a conservative boundary condition and therefore increases our confidence in our predictions of Oldest Ice candidate sites.

(2) The vertical flow that is used in the advection term is based on the equation (2), which is based on Pattyn, 2010 and Van Liefferinge and Pattyn, 2013, but this is certainly an approximation. Please discuss the effect of the assumption that is made on the result in the discussion section. For example, Parrenin et al, 2017 use different assumption but how does the difference affect the present work?
This is a good point. We did not account for the temperature dependence of the flow parameter in Glen's flow law while deriving profiles of vertical velocity. Different shapes of the vertical velocity profile will give rise to differences in the temperature profile, which has not been tested, but has a limited effect on the transient evolution. Similar to the temperature

dependence of both heat capacity and thermal conductivity (neglected in our model), our approximation adds further uncertainty, but as many studies have demonstrated, the uncertainty in geothermal heat flux remains the dominating factor in the transient evolution of ice thermodynamics.

(3) Neglecting the rock temperature calculation would overestimate the amplitude between the glacial-interglacial condition and then can underestimate the GHF limit for the oldest ice. Please discuss this problem in the discussion section.

Most methods calculate average GHF values within or at the surface of the crust, without accounting for gradients of GHF within the crust. From an ice-sheet modelling perspective, it is more realistic to know the temperature gradient at the ice-bed interface rather than a specific GHF value at the interface. The thermal gradient inside the bedrock has an impact on heat availability to the ice (Lowrie, 2007) as does the thermal inertia of the bedrock. Ritz (1987) shows that bedrock temperature will reach equilibrium after thousands of years, on the scale of several climatic cycles, after a change in ice surface temperature. She shows that the use of a 2 km thick crust for the calculation of the crustal thermal gradient is enough to accurately model the changes induced by surface temperature variations. For climate cycles with a 100 kyr cyclicity, the basal temperature perturbation at the bed is ~40% of the surface temperature perturbation if crustal thickness isn't taken into account, and 15% if it is. However knowing the composition, the thickness and the thermal conductivity of the bedrock is also a challenge. At first approximation, we can use a GHF value without taking into account crustal thickness. This simplification is frequently made in glacier and ice sheet models (e.g. Huybrechts et al., 1996; Ritz et al., 1997; Huybrechts and de Wolde, 1999; Pattyn, 2003; Pollard et al., 2005), particularly in steady-state.

We have modified our discussion to describe this simplification explicitly. The paragraph "Knowledge of GHF values at the ice-bedrock interface is a crucial boundary condition for ice flow modelling, yet it remains the most difficult parameter to measure in-situ. Constraining this parameter is therefore essential. GHF is determined by the geology of the bedrock (type of rock, presence of volcanism, etc). However, bedrock geology is unknown in the Antarctic interior and therefore cannot be taken into account in our model" is changed to "From an ice-sheet modelling perspective, it is more realistic to know the temperature gradient at the ice-bed interface rather than a specific GHF value at the interface. The thermal gradient inside the bedrock has an impact on heat availability to the ice (Lowrie, 2007) as does the thermal inertia of the bedrock. Ritz (1987) shows that bedrock temperature will reach equilibrium after thousands of years, on the scale of several climatic cycles, after a change in ice surface temperature. However knowing the composition, the thickness and the thermal conductivity of the bedrock is also a challenge. At first approximation, we can use a GHF value without taking into account crustal thickness. This simplification is frequently made in glacier and ice sheet models (e.g. Huybrechts et al., 1996; Ritz et al., 1997; Huybrechts and de Wolde, 1999; Pattyn, 2003; Pollard et al., 2005), particularly in steady-state."

(4) Why is the threshold of ice thickness to find the oldest ice for Dome F and Dome C different, 2000m and 2500m, respectively? The main reason is that at both sites, the most influential parameter is different. At Dome F, the ice thickness has the strongest influence to keep a sufficient ice-age resolution. Because the ice thickness is, in general, less thick at Dome F than at Dome C with areas with an ice thickness lower than 2500 m. It's for that reason that we used two ice thickness thresholds in the Fig. 7 D: 2000m and 2500m, to show the influence of ice thickness. At Dome C the critical point (among many others) is the bed roughness. In that region, the central area is always thicker than 2500 m. See minor comments further down.

(5) How was the lower limit of ice thickness to find the oldest ice determined?

We used the summary paper of Fisher et al. (2013). This paper discusses ice thickness in detail and shows that 2500 m is a threshold value to keep the bed frozen and to keep a sufficient ice-age resolution. See minor comments as well.

(6) For Dome C, Parrenin et al, 2017 show the location of melt and estimate the possible GHF from modeling and radar data analysis. Discuss what we learn from the present study after knowing the publication of Parrenin et al 2017.
This is a good suggestion. We have added a sentence to state that our results agree well with those of Parrenin et al. (2017), as well as those of Passalcqua et al. (2018), but that because of big differences in the respective study scales of this study with those of Parrenin et al. and Passalcqua et al., we do not go into details.
We have added the following sentence in our Discussion section:
Our results generally agree with those of Parrenin et al. (2017) and Passalacqua et al. (2018). However, because of the difference in spatial scales, a more detailed comparison is beyond the scope of this paper.

(6) Plot the "Little Dome" and Dome C in the map (or show both locations) and discuss the result related to this area.
We agree this would be useful for the readers and have added Little Dome C on all panels of Fig.8 as "LDC" like in Parrenin et al., 2017. We have also added the following in the caption:
LDC locates the Little Dome C area as defined by Parrenin et al., 2017; Cavitte et al.,2018.

minor comments:

P.3 L10-12: refer to the data of GHF known at those cites.
We have now added the relevant citations.
i.e. Vostok (Petit et al., 1999; Parrenin et al., 2004), EPICA Dome C (EPICA community members, 2004; Parrenin et al., 2007), Dome Fuji (Fujii et al., 2002; Hondoh et al., 2002; Watanabe et al., 2003), and EPICA Dronning Maud Land (EPICA community members, 2006; Ruth et al., 2007).

P.3 L17-18, I cannot understand what you mean by this sentence, "Furthermore, the mechanisms that control the geometry …..".
We agree this was unclear and have added the following explanations: Furthermore, the mechanisms that control the geometry and the ice volume as well as Antarctic Ice Sheet stability are also increasingly better understood (Shakun et al., 2015; Pollard et al., 2015). Shakun et al. (2015) put forward the strong coupling between ice volume and temperature over climatic cycles from planktonic 18O records. Pollard et al. (2015) put forward new mechanisms of hydrofracturing and ice cliff failure producing a rapid retreat of the ice sheet during past warm periods.

P.4 Figure1: please show the temperature change of Dome F, too.
As suggested, we have added the temperature variations at Dome F on panel e of Fig.1. However, because of the scale used in this figure, the differences between Dome C and Dome F are too small to be visible. They are only visible if we zoom in on the curve:

[Figure]

P.6 Please show the map of Ts obs (the observed surface temperature) including Dome C and Dome F. It is not clear which dataset is used.

(See comment nr 1)

We noticed that it was not clear where the different datasets came from. We now clearly state where the data comes from. In consequence we changed paragraph 2.3 (Model forcing): Atmospheric forcing is applied in a parameterized way, based on the observed fields of surface mass balance (accumulation rate) based on the output of the RACMO regional atmospheric climate model over the period 1980-2004, calibrated with observed surface mass balance rates (van de Berg et al., 2006; van den Broeke et al., 2006) and surface temperature (van den Broeke, 2008).

We choose not to show the map of T s obs as this can be found easily in the cited papers.

P.6 Ice thickness history taken from Pollard and Deconte, 2009 should be shown (perhaps in Figure 1) for the aid of understanding the results.

We agree with the reviewer. We added panel f showing ice thickness variations at Dome C from Pollard and Deconto (2009) and changed the caption accordingly.

Bottom: f, Ice thickness (m) reconstruction from (Pollard and DeConto, 2009) near Dome C.

[Figure]

P.8 L12, "adding ..basal roughness threshold value of 20m: : :.". The meaning is not clear.

In our paper, we talk about roughness even if, strictly speaking, the roughness represents the bedrock variability on smaller horizontal scales, from millimetres to a few hundred meters (Shepard et al., 2001). This is detailed earlier in the section "Constraints on Oldest Ice candidate sites", which explains that basal roughness represents the standard deviation of the bed topography for the whole area (5km by 5 km).  We added: ",which implies a relatively smooth bed topography," for ease of reading.

P.9 Figure3. Is this for Dome C? Please show both Dome C and Dome F and explain the difference.

This figure is for Dome C. This figure is shown as an example to understand the influence of ice thickness and temperature variations in our calculations. The exact location is not relevant. What is more interesting are the spatial results in Figure 4. This figure is purely for the readers to understand the model output. We agree this was unclear and now mention this explicitly in the caption: Example model result for a location near Dome C. The polynomial fit (black line) indicates the value of GHF needed to keep the bed frozen (corrected for pressure melting).

P.9 L8, "mainly due to shallower ice": how much caused by the ice thickness and surface temperature?.

You are certainly right the lower surface temperature at the Gamburtsev area may also play a role in the high value of the Gpmp. However the influence of the surface temperature is small compared to the influence of ice thickness variations. However, it is difficult to quantify the proportion of the influence of each and so we now state: The Gamburtsev mountains area differs markedly from other regions, with a high Gpmp, between 70 and 100 mW m$^{-2}$ (due to thinner ice cover and lower surface temperatures than at Dome C and Dome Fuji), while the Vostok region presents the lowest Gpmp values.

P. 10 Figure 4. Show in the caption that the area with ice flow within 2 m/yr is displayed. Changed: we added: The green line outlines areas with surface velocities < 2m a-1 (calculated from balance velocities, Pattyn, 2010)

P. 11 L2-3, very good that the higher spatial resolution is shown and discussed in the following section, but the Figure 7 and Figure 8 should be displayed in the same resolution.
We agree that comparing the figures on the same scale would be preferable. However, the Dome Fuji map represents a region that is 400 km wide, while the Dome C region is 200 km wide. We cannot realistically show the two regions on the same scale. However, despite this, both figures have the same model resolution (500 m by 500 m). This is why we avoid direct comparisons of the different sites, but rather list the necessary criteria for the successful recovery of a 1.5 million-year-ice core.

P.11, L7, show the latitude and longitude of "Little Dome C".
We added the distance to Dome C, however this area does not have a very precise location in term of latitude and longitude.
The subglacial highlands 40 km south-west of Dome C, informally named ``Little Dome C'',

P. 12 Why is the red area around the triangle larger in Fig.7 (b) than Fig. 8 (b)? I cannot understand how this was determined.
This is perhaps not clearly stated. In section "Constraints on GHF", we now mention that "the margin of the influence area is 20 km or, if known, the size of the subglacial lake (particularly relevant for the 54 mapped Dome C survey lakes, Young et al., 2017)".

P.12 Figure 7. (d) displays the ice thickness but the Figure 8. (d) displays the Basal topography. For the readers' understanding, it is better to use the common variable (either ice thickness or basal topography). We agree it is preferable to keep the same basemaps but, as described in the general comments above, the most influential parameter on preserving Oldest Ice for two regions differs (see comment above for details). We have now added in the Discussion: We do not show the same base maps for both data sets because ice thickness has the strongest influence on our model results at Dome Fuji while the bed roughness is more relevant for Dome C due to the differences in spatial radar data resolution (Fig . 7D and Fig. 8D).

P.12 L7 "lower bed roughness at Dome Fuji than Dome C" is not easy to understand.
We changed the sentence to: The basal topography (Fig. 7 and Fig. 8) shows a smoother bed (lower bed roughness) at Dome Fuji than at Dome C, enhanced by the difference in radar data cover density.

P.13 Figure8 (a) and (b): Why are the number of triangles and their location different in the two figures?
In Fig. 8 A we put only the Young (2017) lakes and Fig. 8 B we put also the lakes from (Smith et al., 2009; Wright and Siegert, 2012). We corrected the figures for consistency.

P. 16 Figure 9 (d) Why does Snyder (2016) show the "map" of surface temperature change? Snyder 2016 is only providing a time series.
Thank you for the remark, our description was confusing. It is the map made using the reconstruction of Snyder (2016) that we extended over the whole ice sheet. We changed the caption to: D: Reconstructed Variation in the amplitude of surface temperature forced by the multi-proxy database and modelling of Snyder (2016).

P. 16 Figure 9 (d): Surface temperature changes between 1.5Ma and 0 Ma? This is not possible. Please check and rewrite what you mean. We agree that the wording is confusing. We changed the sentence to "Reconstructed variation in the amplitude of temperature"

P.17 Figure 10: Clarify which difference (and the sign) is meant.
We clarified the caption as follows: Comparison of the GHF needed to reach the pressure melting point calculated using the Van Liefferinge and Pattyn (2013) simple model and the Gpmp calculated in this paper (mW m-2). Blue colours (negative values) indicate that we need a higher GHF for the Gpmp to reach the pressure melting point, and vice versa for the positive values (in red).

P.17 L6, explain more why " less promising due to thinner ice cover". Thinner ice could be promising in freezing condition. Discuss the advantage and disadvantage.
We of course agree that a thinner ice cover can lead to a higher probability of having a frozen bed. However, as explained in Fischer et al,. (2013), we also need to have a sufficient age resolution at the base ("*At the same time the ice may not be too thin to find old ice sufficiently above bedrock*") We therefore changed the sentence to: Plateau areas also show potential Oldest Ice sites, but these are less promising due to their lower age resolution as a result of their thinner ice cover.

P.17 L11: Where are the "two areas"? See response P.18 L4. We added a summary figure.

P.17 L13: "evocative of a horst" is not understandable.
We agree that this is perhaps too jargon-y to end with. We extended our description as follows: The geometry of this elongated bedrock feature is evocative of a raised fault block (also referred to a horst in geology) which, if confirmed, implies an uplifted but relatively flat bedrock surface.

P.18 L4: "a number of candidate locations" are not clear in the figures. Please make a summary figure to enlarge and focus the locations.
Thank you for your suggestion. We added a separate figure (Fig.10) focusing on the Little Dome C and Dome C regions. Following a discussion with Catherine Ritz we also added in this figure the promising OI sites using exactly the same parameter values except for a bed roughness (standard deviation of the bed elevation) of 30 m instead of 20 m.  We can see for Little Dome C region the influence of the bed roughness.

[Figure]

Figure 11. Detail of promising sites around Little Dome C. Potential locations of Oldest Ice are shown in red for H > 2500 m, sigma b < 20 m, a probability that ice has reached the pressure melting point less than 0.4 and a distance to radar lines less than 2 km.  In pink, potential locations of Oldest Ice with the same parameter values but  a value of sigma b < 30 m. The background displays basal topography from Young et al. (2017). The blue dashed line locates the "elongated bedrock feature" discussed in

the paper. The green dashed lines locate the 2015/2016 radar survey. LDC and the dashed rectangle locate the Little Dome C area as defined by Parrenin et al. (2017) and Cavitte et al. (2018).

We added in the caption of the Fig.8: Fig. 11. is a detailed view of the Little Dome C region.